# Advanced Flow Cytometry Using the SYTO-13 Dye for the Assessment of Platelet Reactivity and Maturity in Whole Blood

**DOI:** 10.3390/mps6010008

**Published:** 2023-01-13

**Authors:** Oliver Buchhave Pedersen, Leonardo Pasalic, Erik Lerkevang Grove, Steen Dalby Kristensen, Anne-Mette Hvas, Peter H. Nissen

**Affiliations:** 1Thrombosis and Haemostasis Research Unit, Department of Clinical Biochemistry, Aarhus University Hospital, 8200 Aarhus, Denmark; 2Department of Cardiology, Aarhus University Hospital, 8200 Aarhus, Denmark; 3Department of Clinical Medicine, Faculty of Health, Aarhus University, 8200 Aarhus, Denmark; 4Institute of Clinical Pathology and Medical Research, Departments of Clinical and Laboratory Haematology, Westmead University Hospital, Sydney 2145, Australia; 5Westmead Clinical School, Faculty of Medicine and Health, University of Sydney, Sydney 2145, Australia; 6Faculty of Health, Aarhus University, 8200 Aarhus, Denmark

**Keywords:** flow cytometry, platelet reactivity, platelet maturity, immature platelets, SYTO-13 dye

## Abstract

Newly produced immature platelets are larger, contain higher amounts of residual RNA, and are more reactive than mature platelets. Flow cytometry using the SYTO-13 dye is a method for the subdivision of immature platelets from mature platelets based on the labelling of intracellular platelet RNA, enabling the simultaneous investigation of the reactivity of each platelet population. This method provides detailed information on several aspects of platelet physiology using a combination of platelet surface markers and agonists. Currently, no standardized protocol exists across laboratories. Here, we describe a flow cytometry protocol in detail to investigate platelet reactivity and its relation to platelet maturity. We analyzed 20 healthy individuals with the protocol and compared the platelet subpopulation with the highest SYTO-13 labelling (in the first quintile, “SYTO-high”) corresponding to the most immature platelets (highest RNA content) with the platelet subpopulation with the lowest SYTO-13 labelling (in the fifth quintile, “SYTO-low”) corresponding to the mature platelets with the lowest RNA content. SYTO-high platelets had overall significantly increased platelet reactivity compared with that of SYTO-low platelets. The presented method may be a valuable research tool for the analysis of platelet reactivity and its relation to platelet maturity.

## 1. Introduction

Flow cytometry can quantify markers of platelet reactivity through the measurements of the expression of platelet surface antigens using specific fluorescence-coupled antibodies [1,2,3]. Many aspects of platelet physiology can be evaluated simultaneously by measuring several surface markers via using a combination of different antibodies and agonists [4]. Flow cytometry can be performed in patients regardless of platelet count, as platelets are analyzed individually [5,6]. Furthermore, flow cytometry only requires small sample volumes and can be performed on whole blood or platelet-rich-plasma [4,7]. However, flow cytometry requires experienced laboratory personnel and advanced equipment, and is currently not standardized across laboratories [8].

Immature platelets are newly produced platelets that are more reactive than mature platelets [9,10,11]. Compared with mature platelets, immature platelets contain larger amounts of RNA derived from megakaryocytes providing the ability to produce proteins that contribute to platelet reactivity [12,13]. RNA degenerates in a time-dependent manner [14]. Strong evidence suggests a positive association between immature platelets and the risk of cardiovascular events in patients with ischemic heart disease [15,16,17,18]. Hence, exploring and characterizing immature platelets is of great clinical interest.

The most common method to quantify immature platelets is immature platelet count (IPC) or the immature platelet fraction of the total platelet pool (IPF). These parameters are most often measured with a fully automated assay using monochromatic automated flow cytometry (e.g., Sysmex^®^). This method uses a predefined algorithm to classify immature platelets on the basis of manufacturer-decided cut-offs for forward-scattered (FSC) light (corresponding to cell size), side-scattered (SSC) light (corresponding to granularity), and the staining of nucleic acids using fluorescent dyes [19]. However, this method does not allow for the further investigation of the reactivity of immature platelets.

The cell-permeable SYTO-13 fluorescent dye exhibits bright green fluorescence upon binding to nucleic acids, including RNA. The amount of SYTO-13 correlates with the quantity of RNA within the platelet and with the number of immature platelets [20]. Hence, SYTO-13 staining using flow cytometry provides a method for subdividing a platelet population into immature and mature platelets, enabling the detailed investigation of immature platelets [20]. In addition, SYTO-13 staining may be advantageous compared with other nucleic acid dyes (e.g., thiazole orange (TO)), as SYTO-13 staining is stable over time and shows better correlation with immature platelets measured with automatic assays compared with TO [20]. Therefore, although more demanding and time-consuming, manual flow cytometry using SYTO-13 staining is a promising method for subdividing immature platelets from mature platelets and subsequently investigating the reactivity of both platelet populations.

The international standardization of advanced laboratory methodology between laboratories is very important. The present protocol describes in detail the experimental design and workflow to examine platelet reactivity and platelet maturity using whole blood flow cytometry.

## 2. Materials

### 2.1. Flow Cytometer

All analyses were performed on a CytoFLEX S (B75442) flow cytometer (Beckman Coulter, Miami, FL, USA) equipped with a violet laser (405 nm), a blue laser (488 nm), a yellow-green laser (561 nm), and a red laser (638 nm), and with 11 detectors.

### 2.2. Blood Sampling

See Section 6.1.

### 2.3. Reagents

CD42b-Alexa Flour 700 (AF700) (stock concentration: 100 µg/mL, mouse antibody, clone HIP1, Nordic BioSite, Copenhagen, Denmark, cat. no. 303928).CD45-Brilliant Violet 650 (BV650) (stock concentration: 100 µg/mL, mouse antibody, clone HI30, Nordic BioSite, Copenhagen, Denmark, cat. no. 304044).CD63-Phycoerythrin-Cyanine 7 (PECy7) (stock concentration: 200 µg/mL, mouse antibody, clone H5C6, Nordic BioSite, Copenhagen, Denmark, cat. no. 353010).CD62p-PE (P-selectin, stock concentration: 100 µg/mL, mouse antibody, clone AK4, Nordic BioSite, Copenhagen, Denmark, cat. no. 304906).Antifibrinogen (stock concentration: 1 mg/mL unconjugated, polyclonal chicken, Diapensia HB, Linköping, Sweden, cat. no. 2204).ReadiLink™ Rapid mFluor™ Violet 420 (V420) Antibody Labeling Kit (Nordic BioSite, Copenhagen, Denmark, cat. no. ABD-1105).SYTO-13 (stock concentration: 5 mM in dimethyl sulfoxide (DMSO), ThermoFisher Scientific, Copenhagen, Denmark, ref. no. S7575).NaCl (137 mmol, Merck, Darmstadt, Germany, ref. no. 1.06404.1000).KCl (2.7 mmol, Merck, Darmstadt, Germany, ref. no. 1.04936.0500).MgCl_2_ (1 mmol, Merck, Darmstadt, Germany, ref. no. 1.05833.0250).Glucose (5.6 mmol, Sigma-Aldrich, St. Louis, MO, USA, ref. no. G7021-100G).4-(2-hydroxyethyl)-1-piper-azineethanesulfonic acid (HEPES) (20 mmol, Sigma-Aldrich, St. Louis, MO, USA, ref. no. H3375-100G).Bovine Serum Albumin (BSA, Sigma-Aldrich, St. Louis, MO, USA, ref. no. A7030.100G).Paraformaldehyde (PFA) (Formaldehyde Solution >36.0%, Sigma-Aldrich, St. Louis, MO, USA, ref. no. 47608-250ML-P).Phosphate-buffered saline (PBS) (one tablet, Sigma-Aldrich, St. Louis, MO, USA, ref. no. P441750TAB).Demineralized water (ELGA Purelab flex, Krüger Aquacare, Glostrup, Denmark).Titriplex (EDTA) (Merck, Darmstadt, Germany, ref. no. 1.08418.0100).PE isotype control (stock concentration: 0.2 mg/mL Mouse IgG1, κ isotype ctrl, Nordic BioSite, Copenhagen, Denmark, cat. no. 400112).Adenosine diphosphate (ADP, stock concentration: 1.4 µM) (Sigma-Aldrich, St. Louis, MO, USA, ref. no A5285-16).Thrombin Receptor Activating Peptide (TRAP-6, stock concentration: 2.6 µM) (JPT, Berlin, Germany, ref. no. 17951-4).Arachidonic Acid (AA, stock concentration: 15.3 mM) (Sigma-Aldrich, St. Louis, MO, USA, ref. no. SML 1395-100 MG).Collagen-related peptide (collagen, stock concentration: 7mg/mL) (CambCOL Ltd, Cambridge, UK, ref. 1 mg CRP-XL freeze-dried).

### 2.4. Equipment

Nunc MaxiSorp™, flat-bottom (ThermoFisher Scientific, Copenhagen, Denmark, cat. no. 44-2404-21).Reaction vials (1.5 mL) (Chromsystems®, Munich, Germany, order no. 33005).Superclear® 12 × 75 mm culture tubes (test tubes) (Labcon, Petaluma, CA, USA, ref. no. 3350-368-000-9).Wizard Advanced IR Vortex Mixer (VELP Scientific Inc, New York, NY, USA).CytoFLEX S (B75442) flow cytometer (In deep well configuration, Beckman Coulter, Miami, FL, USA).CytExpert Software (Version 2,4, Beckman Coulter, Miami, FL, USA).

## 3. Procedure

### 3.1. Storage of Reagents

HEPES buffer was collected from the freezer or cooler. See Section 6.2.Antibodies and PE isotypic control were collected from the cooler.
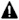
**CRITICAL STEP:** HEPES buffer, antibodies, and PE isotype control should reach room temperature before further processing.

### 3.2. Preparation of Reagents

1.Antibodies and isotypic control dilution were prepared in seven labelled reaction vials.2.P-selectin-PE: 1 µL antibody in 74 µL HEPES (dilution factor 1:75).3.CD63-PCy7: 25 µL antibody in 25 µL HEPES (dilution factor 1:2).4.Antifibrinogen-V420 (See Section 6.3): 25 µL antibody in 25 µL HEPES (dilution factor 1:2).5.SYTO-13: 1 µL dye in 19 µL HEPES (dilution factor 1:20).6.PE isotype control: 1 µL antibody in 150 µL HEPES (dilution factor 1:150).7.CD42b-AF700: undiluted.8.CD45-BV650: undiluted.9.An antibody cocktail was prepared in a single reaction vial using the previously diluted antibodies/reagents: 32 µL P-selectin-PE, 32 µL CD63-PCy7, 32 µL Antifibrinogen-V420, 32 µL SYTO-13, 48 µL CD42b-AF700, 3.2 µL CD45-BV650 and 172.8 µL HEPES buffer.

### 3.3. Preparations of Working Dilutions of Agonists

10.Agonists were collected from the freezer.11.
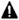
**CRITICAL STEP:** keep agonists in the freezer until immediately before use, thaw at room temperature for 5 min.12.Agonist dilution was prepared in Eppendorf tubes.13.Collagen:14.Add 1 µL agonist in 99 µL HEPES buffer (solution A).15.Add 1 µL of solution A in 99 µL HEPES buffer (working concentration: 0.7 μg/mL).16.ADP: Add 5 µL of agonist in 45 µL HEPES buffer (working concentration: 140 μM).17.TRAP: Add 10 µL of agonist in 60 µL HEPES buffer (working concentration: 371 μM).18.AA: Add 15 µL of agonist in 15.6 µL HEPES buffer (working concentration: 7.5 mM).

### 3.4. Preparing Test Tubes

19.Six test tubes were prepared:20.Negative/isotype sample: 5 µL CD63-PCy7 (diluted), 5 µL antifibrinogen-V420 (diluted), 7.5 µL CD42b-AF700 (undiluted), 0.5 CD45-BV650 (undiluted), 5 µL PE isotype control, 5 µL EDTA (undiluted) and 32 µL HEPES buffer.21.HEPES/preactivation sample: 55 µL antibody cocktail and 5 µL HEPES buffer.22.Collagen sample: 55 µL antibody cocktail and 5 µL collagen (final assay concentration: 0.05 μg/mL).23.ADP sample: 55 µL antibody cocktail and 5 µL ADP (final assay concentration: 10.8 μM).24.TRAP sample: 55 µL antibody cocktail and 5 µL TRAP (final assay concentration: 28.6 μM).25.AA sample: 55 µL antibody cocktail and 5 µL AA (final assay concentration: 0.58 μM).26.
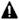
**CRITICAL STEP:** antibodies and PE isotype control must not be exposed to direct light for a prolonged period.27.A total of 5 µL whole blood was added to each test tube at 20 s intervals.28.Each sample was mixed slightly by gently shaking the tube.29.
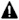
**CRITICAL STEP:** incubate for exactly 10 min in darkness.30.After incubation, 2 mL PBS-0.2% PFA was added to each test tube for fixation with 20 s intervals. See Section 6.4.31.
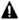
**CRITICAL STEP:** after adding fixative, each test tubes must be vortexed immediately for 8–10 s.32.After fixation, the test tubes must rest in darkness for 1 h.

### 3.5. Preparing the Analysis

33.The waste bottle was emptied and refilled with sheath fluid container.34.CytoFLEX and the associated computer were started.35.CytExpert software was started.36.The system startup program was run: see Section 6.6.37.The daily quality control program was run: see Section 6.7.38.A new experiment was created from the template: see Section 6.8.

### 3.6. Analysis (This May Take up to 90 min Depending on Platelet Count)

39.A well plate was prepared.40.Using a manual pipette, add a sample of Test Tube 1 into Well 1, Test Tube 2 into Well 2, and so forth.41.Inset well plate onto plate loader on CytoFLEX.42.Load the plate.43.Press Initialize in CytExpert software.44.Press Backflush in CytExpert software.45.Press Autorecord in CytExpert software.46.
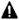
**CRITICAL STEP:** the placement of the test wells in CytExpert software must match the added wells on the plate.

### 3.7. Data Quality

47.Preactivation was evaluated on the basis of P-selectin expression in Test Tube 2 containing the antibody cocktail, HEPES buffer, and whole blood (see Section 4.2).48.P-selectin expression < 15% was considered to be acceptable (no evidence of preactivation).49.In the case of P-selectin expression > 15%, the results should be evaluated with caution.

## 4. Results and Interpretation

### 4.1. Identifying Single Platelets

50.Single platelets were identified for further analysis using the gating strategy as shown in Figure 1. Minor adjustments are to be expected between test tubes; see Section 6.9.

### 4.2. Defining Gates and Measurements

51.The expression of activation-dependent platelet surface markers (antifibrinogen, CD63, and P-selectin) was evaluated via the percentage of positive platelets and with median fluorescence intensity (MFI) corresponding to the amount of the expression of a surface marker.52.Test Tube 1 (negative sample) was used to set the positive gates, as shown in Figure 2A. The positive gates were set to include 1–2% events for antifibrinogen and CD63, 0.1–0.2% events for P-selectin, and 0.02–0.04% events for SYTO-13 on the negative control. Gates must be applied to all tubes, although minor adjustment can be expected between test tubes. Preactivation was evaluated through the positive expression of P-selectin in Test Tube 2 containing the antibody cocktail, whole blood, and HEPES buffer instead of an agonist, as shown in Figure 2B. Furthermore, using the HEPES sample, all single platelets were subdivided into quintiles according to SYTO MFI. SYTO-high platelets occupied the first quintile (20% of platelets with the highest SYTO-13 expression) and SYTO-low platelets occupied the fifth quintile (20% of the platelets with the lowest SYTO-13 expression), as shown in Figure 2C. In Test Tubes 3–6, containing agonists, the MFI and percentage of positive platelets of activation-dependent platelet surface markers of all single platelet were evaluated as shown in Figure 2D, SYTO-high platelets in Figure 2E, and SYTO-low platelets in Figure 2F.

### 4.3. Results

53.Table 1 shows the flow cytometric analysis of platelet reactivity in 20 healthy individuals investigated using the current protocol. The expression of antifibrinogen indicates platelet-to-platelet aggregation, as this process is mediated through fibrinogen binding [21]. The expression of CD63 indicates the release of dense granules containing high concentrations of ADP that increase platelet reactivity [1]. The expression of P-selectin indicates the release of α-granules containing proteins participating in platelet reactivity, cell adhesion, and coagulation [1].

54.Table 2 shows the platelet reactivity of SYTO-high platelets (immature platelets) and SYTO-low platelets (mature platelets) in 20 healthy individuals investigated via flow cytometry using the described protocol. SYTO-high platelets were statistically significantly larger (higher FSC) than the SYTO-13-low platelets. Furthermore, SYTO-high platelets had overall statistically significantly increased platelet reactivity compared with the SYTO-low platelets.

Figure 3 shows platelet reactivity in 20 healthy individuals in all single platelets, subdivided into SYTO-high platelets (immature platelets) and SYTO-low platelets (mature platelets). Overall, SYTO-high platelets had the highest platelet reactivity, and SYTO-low platelets had the lowest platelet reactivity.

## 5. Summary and Conclusions

We provided a protocol to investigate how platelet reactivity is related to platelet maturity in whole blood. Platelet reactivity was measured via the expression of antifibrinogen, CD63, and P-selectin. Furthermore, all single platelets were subdivided on the basis of SYTO-13 labelling into 20% SYTO-high platelets (immature platelets) and 20% SYTO-low platelets (mature platelets) corresponding to the RNA content within the platelets. Simultaneously, the platelet reactivity of each platelet subpopulation was measured. 

This protocol requires experienced laboratory personnel and advanced equipment. Currently, the method has not been standardized across laboratories. These factors may limit implementation into routine clinical practice. However, the method provides a valuable research tool to analyze platelet reactivity and its relation to platelet maturity.

## 6. Notes

### 6.1. Blood Sampling

Whole blood was collected in tubes anticoagulated with 3.2% sodium citrate (Terumo Europe, Leuven, Belgium) using a large needle (21-gauge) with a minimum of stasis. Samples were then allowed to rest for 1 h before any sample preparation and fixation as previously described [5].

### 6.2. HEPES Buffer

HEPES buffer was prepared by mixing the following: 4.003 g NaCl, 0.101 g KCl, 0.103 g MgCl_2_, 0.504 g glucose, 2.383 g HEPES, and 0.5 g BSA. Then, sterile water was added until a total volume of 500 mL. Lastly, the pH of the HEPES buffer was adjusted to 7.4, and it was stored at –20°C until use.

### 6.3. Conjugation of Antifibrinogen with V420

Antifibrinogen was conjugated inhouse using the ReadiLink™ Rapid mFluor™ V420 Antibody Labeling Kit (Nordic BioSite, Copenhagen, Denmark) following the manufacturer’s instructions.

### 6.4. Mixture of PBS-0.2% PFA Fixation

One tablet of PBS was diluted in 200 mL demineralized water. Then, 240 µL 37% PFA was added to 44.4 mL PBS. The fixation buffer was then ready for use.

### 6.5. Flow Cytometer Settings

All analyses were performed with the corresponding filter configurations for the fluorescence detectors, as shown in Table 3. An unstained sample was used to adjust the gain for all fluorochromes to maximize the separation of positive and negative events. Settings were confirmed by matching the isotypic controls. The sampling rate was set to slow to minimize platelet activation. For each test well, a minimum of 10,000–15,000 platelets or a total sample time of 600 sec was assessed.

### 6.6. System Startup Program

Before analyzing the samples, different startup programs were run using the incorporated programs according to manufacturer’s instructions.

### 6.7. Daily Quality Control Program

The quality control of fluorescence and particle size, and the alignment of the optical and fluidic systems were standardized and controlled with QC beads (CytoFLEX Daily QC Fluorophores, REF B53230, Beckman Coulter, Miami, FL, USA) daily according to the manufacturer’s instructions.

### 6.8. Experiment Creation

“New Experiment from Template” was selected on the Start page.“Browse” was selected to save the experiment in the preferred location.“Template” was selected to choose the specific template (a template must be created before analysis).

### 6.9. Identification of Platelets

Platelets were identified via forward-scatter (FSC) and sideward-scatter (SSC) characteristics, and using specific platelet marker CD42b. Additionally, CD45 (a specific leucocyte marker) was used to exclude single leucocytes and platelet–leucocyte aggregates, thereby ensuring a pure platelet population. Lastly, by employing a FSC peak and FSC intensity plot, microparticles and platelet–platelet aggregates were excluded. Hence, single platelets were included for further analysis. This identification strategy was employed in all further experiments. The volume and concentration of CD42b and CD45 were selected to ensure the optimal separation of negative and positive events.

### 6.10. Stability of SYTO-13 Dye

The stability of SYTO-13 staining in whole blood was investigated for a total of 3 h comparing incubation times of 10 and 30 min.
A total of 5 µL of SYTO-13 with a final concentration of 13.7 µM, 7.5 µL of CD42b, 0.5 µL of CD45, and 47.0 µL HEPES was pooled.Then, 5 µL whole blood was added.The sample was incubated in darkness for 10 (Figure 4) or 30 min (Figure 5).Fixation with 2 mL of 0.2% PFA-PBS.After fixation, samples were transferred to the well plate and analyzed.The analysis started by running an unstained sample, analyzed in 6 min, followed by the analysis of a stained sample for 4 min.The unstained sample was used to set the gate to subdivide platelets into SYTO-13-positive and SYTO-13 negative.Steps 6 and 7 were repeated for a total of 3 h.The MFI and percentage of positive platelets were stable after 10 min of incubation followed by 60 min of resting. On the basis of these findings, we decided to use these time intervals in our future analyses.

### 6.11. Titration of Antibodies

To find the optimal concentration of SYTO-13, antibody, and activation-dependent surface markers, titration experiments were performed using whole blood resting for 60 min prior to sample preparation, incubation time of 10 min followed by fixation, and 60 min of resting in the dark. Furthermore, the selected concentrations were controlled with matching isotypic controls with similar concentrations showing minimal nonspecific binding.
A final SYTO-13 concentration of 19.23 µM was selected.A final P-selectin concentration of 0.10 µg/mL was selected.A final CD63 concentration of 7.69 µg/mL was selected.A final antifibrinogen concentration of 0.013 µg/mL was selected.

We experienced considerable lot-to-lot variation in antibodies. Antibody titration is, therefore, crucial when changing lot numbers to achieve the same percentage of positive platelets and MFI values.

### 6.12. Stability of Activation Dependent Surface Markers

We examined the stability of the activation-dependent surface markers to investigate if the expression of surface markers was stable over a period of 60 min after fixation (selected as the expression of SYTO-13 that was stable after 60 min of resting), as shown in Table 4. Overall, our results show no significant difference in the expression of activation-dependent surface markers between the two time points. In the final protocol, we decided to include a 60 min rest period after incubation, as both SYTO-13 and the activation-dependent surface markers were stable after 60 min.

### 6.13. Titration of Agonists

All agonists were titrated to saturating concentration as previously described [5].

### 6.14. Agonist Dilution

All agonist were diluted in accordance with Table 5.

### 6.15. Compensation

To minimize spectral overlap, a compensation matrix was created using CytExpert software and the designated compensation in accordance with the manufacturer’s instructions. By using this template, compensation values were automatically calculated using single-stained activated platelets for each color, as shown in Table 6.

## Figures and Tables

**Figure 1 mps-06-00008-f001:**
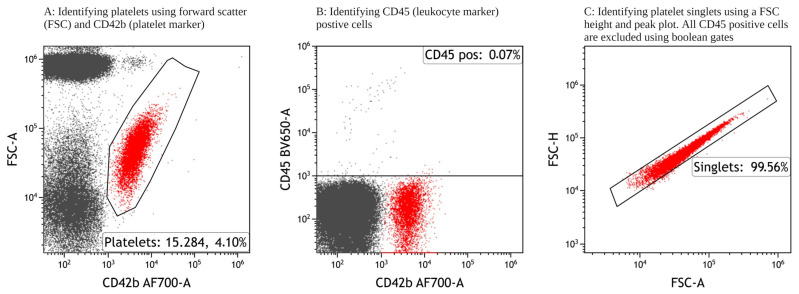
Gating strategy to identify single platelets by excluding CD45-positive cells (leukocytes), platelet-platelet aggregation, platelet-leukocyte aggregation, and microparticles. FSC-A: forward scatter area, FSC-H: forward scatter height.

**Figure 2 mps-06-00008-f002:**
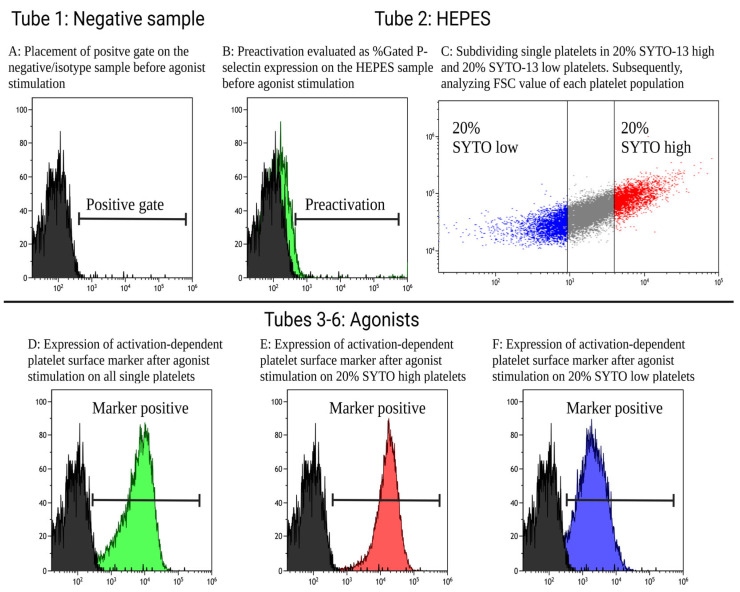
Workflow illustrating (**A**) the placement of the positive gate in Test Tube 1, (**B**) the evaluation of preactivation in Test Tube 2, (**C**) subdividing platelets in SYTO-13-high and -low platelets in Test Tube 2, and the measurement of activation-dependent platelet surface markers after agonist stimulation in Test Tubes 3–6 in (**D**) all single platelets, (**E**) SYTO-13 high platelets, and (**F**) SYTO-13 low platelets.

**Figure 3 mps-06-00008-f003:**
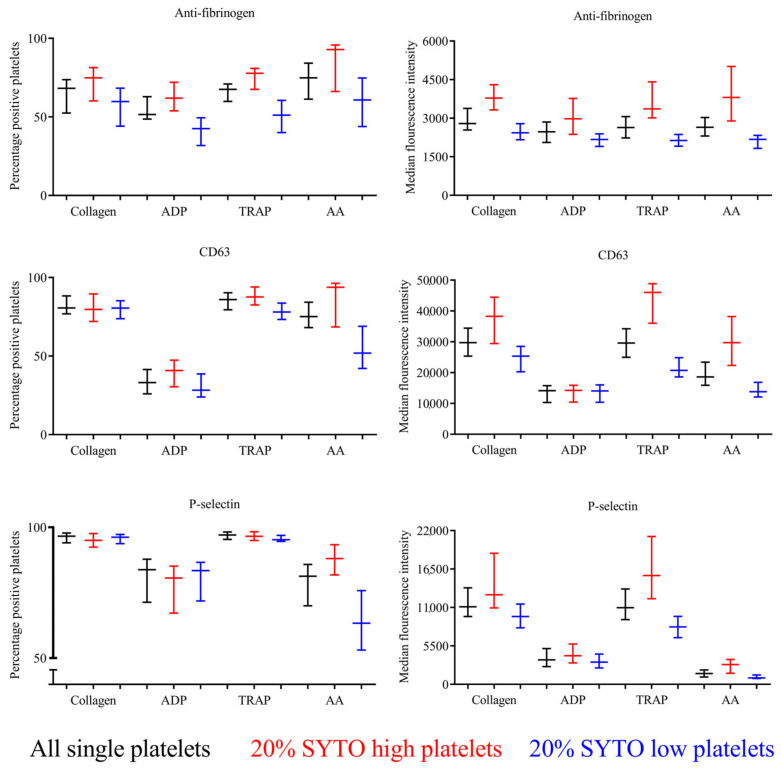
Percentage of positive platelets and median fluorescence intensity of activation-dependent surface markers after agonist stimulation divided into all single platelets (black), SYTO-high platelets (red), and SYTO-low platelets (blue). All values are presented as median and interquartile range. Collagen: collagen-related-peptide, ADP: adenosine diphosphate, TRAP: thrombin-receptor-activating-peptide, AA, arachidonic acid. Median and interquartile range are presented.

**Figure 4 mps-06-00008-f004:**
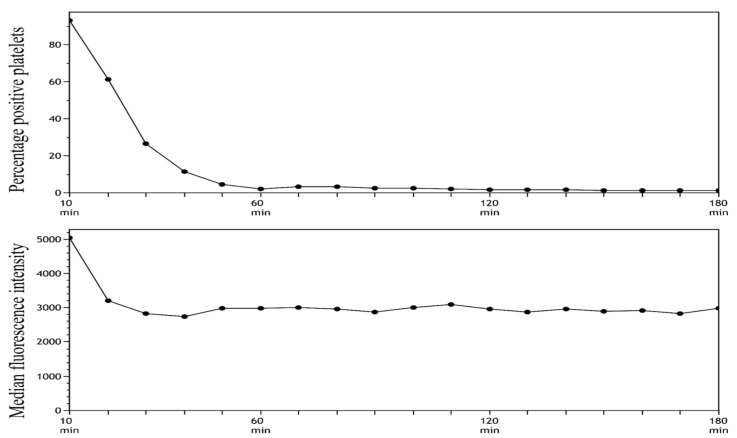
Stability of SYTO-13 staining in whole blood after 10 min of incubation analyses for a total of 3 h (180 min).

**Figure 5 mps-06-00008-f005:**
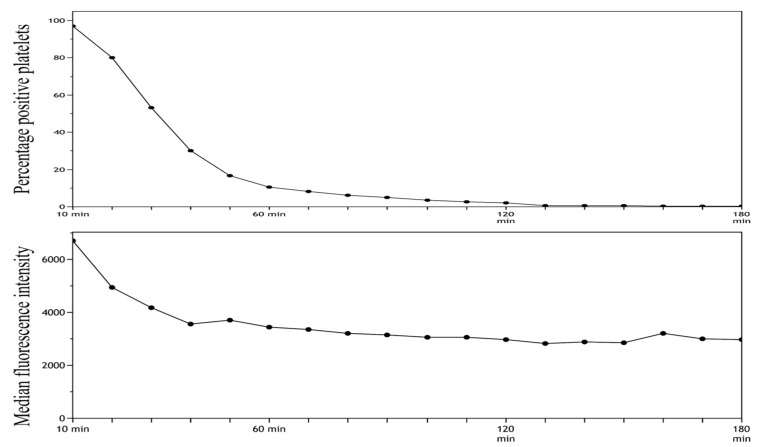
Stability of SYTO-13 staining in whole blood after 30 min of incubation analyses for a total of 3 h (180 min).

**Table 1 mps-06-00008-t001:** Analysis of agonist-induced platelet reactivity surface markers in 20 healthy individuals using flow cytometry.

Reactivity Markers	Percentage of Positive Platelets	MFI
Antifib (COL)	68 (53;74)	2791 (2540;3383)
CD63 (COL)	81 (77;88)	29,746 (25,375;34,419)
P-sel (COL)	97 (94;98)	11,095 (9703;13,817)
Antifib (ADP)	52 (49;63)	2474 (2056;2852)
CD63 (ADP)	33 (26;42)	14,148 (10,332;15,784)
P-sel (ADP)	84 (71;88)	3504 (2545;5127)
Antifib (TRAP)	68 (60;71)	2638 (2232;3060)
CD63 (TRAP)	86 (80;90)	29,620 (24,975;34,233)
P-sel (TRAP)	97 (95;98)	10,967 (9266;13,644)
Antifib AA	75 (61;84)	2648 (2309;3030)
CD63 (AA)	75 (68;84)	18,610 (15,891;23,403)
P-sel (AA)	81 (70;86)	1542 (1051;2064)
Preactivation (P-sel expression in resting platelets)	2 (2;3)	

All values are presented as median and interquartile range. Abbreviations: MFI: median fluorescence intensity, Antifib: antifibrinogen, P-sel: P-selectin, COL: collagen-related peptide, ADP: adenosine diphosphate, TRAP: thrombin-receptor-activating-peptide, AA, arachidonic acid.

**Table 2 mps-06-00008-t002:** Platelet reactivity of 20% SYTO-13-high platelets and 20% SYTO-13-low platelets in healthy individuals investigated with flow cytometry.

	SYTO-High Platelets (Immature Platelets)		SYTO-Low Platelets (Mature Platelets)		*p*-Value of Difference (High vs. Low)	
Markers	MFI		MFI			
FSC (HEPES)	84,934		30035		<0.0001	
	75,527;94,029		25,376;31,368			
**Reactivity Markers**	**Percentage of positive platelets**	**MFI**	**Percentage of positive platelets**	**MFI**	**Percentage of positive platelets**	**MFI**
Antifib (COL)	75	3782	60	2433	**<0.0001**	**<0.0001**
	60;81	3324;4305	44;68	2160;2785		
CD63 (COL)	80	38,284	81	25,362	0.50	**<0.0001**
	72;90	29,456;44,463	74;85	20,275;28,528		
P-sel (COL)	95	12,822	96	9699	0.09	**<0.0001**
	92;98	10,938;18,772	94;97	8086;11,486		
Antifib (ADP)	62	2975	43	2172	**<0.0001**	**<0.0001**
	54;72	2373;3769	32;50	1904;2390		
CD63 (ADP)	41	14,233	29	14,075	**<0.0001**	0.73
	31;47	10,424;15,893	24;39	10,383;16,021		
P-sel (ADP)	81	4096	84	3172	0.51	**<0.0001**
	67;85	3071;5773	72;87	2352;4334		
Antifib (TRAP)	78	3365	51	2133	**<0.0001**	**<0.0001**
	68;81	3012;4412	40;61	1911;2368		
CD63 (TRAP)	88	46,033	78	20,717	**<0.0001**	**<0.0001**
	83;94	36,010;48,869	73;84	18,589;24,851		
P-sel (TRAP)	97	15,579	95	8218	**0.03**	**<0.0001**
	95;98	12,256;21,171	95;97	6681;9734		
Antifib (AA)	93	3809	61	2177	**0.0003**	**<0.0001**
	66;96	2894;5013	44;75	1828;2334		
CD63 (AA)	94	29,713	52	13,829	**0.0003**	**<0.0001**
	69;96	22,360;38,195	42;69	12,101;16,868		
P-sel (AA)	88	2838	63	927	**0.0005**	**<0.0001**
	82;93	1583;3567	53;76	842;1340		

All values are presented as median and interquartile range. Differences were analyzed using the Mann–Whitney test. Test significance was two-tailed with a probability value of *p* < 0.05. All significant *p*-values are highlighted with bold. Abbreviations: MFI: median fluorescence intensity, FSC: forward scatter, Antifib: antifibrinogen, P-sel: P-selectin, COL: collagen-related peptide, ADP: adenosine diphosphate, TRAP: thrombin-receptor-activating peptide, AA, arachidonic acid.

**Table 3 mps-06-00008-t003:** Antibodies, fluorescence, and corresponding flow cytometer settings.

Antibodies	Fluorescence	Channel	Laser	Filter
CD42b	AF700	FL3	Red	712/25
CD45	BV650	FL5	Violet	780/60
Antifibrinogen	V420	FL4	Violet	450/45
CD63	PECy7	FL7	Yellow-Green	780/60
CD62p/P-selectin	PE	FL6	Yellow-Green	585/42
SYTO-13		FL1	Blue	525/40

Abbreviations: AF700: Alexa Flour 700, BV650: Brilliant Violet 650, V420: Violet 420, PECy7: Phycoerythrin-Cyanine 7, PE: Phycoerythrin.

**Table 4 mps-06-00008-t004:** Comparing the expression of activation-dependent platelet surface analyzed directly after fixation and analyzed after 60 min of resting after fixation.

	Marker		Directly after Fixation	60 min Resting after Fixation	Percentage Difference
HEPES	Preactivation (P-selectin)	% positive	3	3	13
		MFI	658	686	4
Collagen	Antifibrinogen	% positive	85	85	0.0
		MFI	2138	2194	3
	CD63	% positive	72	74	4
		MFI	28,925	29,086	1
	P-selectin	% positive	91	9	0
		MFI	9225	9334	1
ADP	Antifibrinogen	% positive	79	80	1
		MFI	2237	2277	2
	CD63	% positive	21	23	8
		MFI	13,758	13,472	2
	P-selectin	% positive	79	77	4
		MFI	2708	2518	7
TRAP	Antifibrinogen	% positive	88	88	0
		MFI	2365	2425	3
	CD63	% positive	87	88	1
		MFI	30,664	31,976	4
	P-selectin	% positive	98	98	0
		MFI	11,594	11,192	3
Arachidonic Acid	Antifibrinogen	% positive	96	95	1
		MFI	2613	2630	1
	CD63	% positive	79	85	7
		MFI	23,678	26,036	10
	P-selectin	% positive	91	88	3
		MFI	2265	1748	23

Abbreviations: ADP: adenosine diphosphate. TRAP: thrombin-receptor-activating-peptide. MFI: median fluorescence intensity. Collagen: collagen-related peptide.

**Table 5 mps-06-00008-t005:** Agonist dilution.

	ADP	TRAP	Collagen	AA
**Agonist stock conc.**	1.4 mM	2.6 µL	7 mg/mL	15.3 mM
**Agonist stock volume**	5 µL	10 µL	1 µL *	15 µL
**HEPES buffer volume**	45 µL	60 µL	99 µL	15.6 µL
**Agonist working dilution volume**	50 µL	70 µL	100 µL	30.6 µL
**Agonist working dilution conc.**	140 µM	371 µM	0.7 µg/mL	7.5 mM
**Blood sample**	5 µL	5 µL	5 µL	5 µL
**Antibody cocktail volume**	55 µL	55 µL	55 µL	55 µL
**Agonist (working dilution) volume**	5 µL	5 µL	5 µL	5 µL
**Final assay volume**	65 µL	65 µL	65 µL	65 µL
**Agonist final assay conc**	10.8 µM	28.6 µM	0.05 µg/mL	0.58 µM

Abbreviations: ADP: adenosine diphosphate. TRAP: thrombin-receptor-activating-peptide, Collagen: collagen-related peptide, conc: concentration. * First, 1 µL agonist was added in 99 µL HEPES buffer (solution A). Then, 1 µL of solution A was added in 99 µL HEPES.

**Table 6 mps-06-00008-t006:** Compensation matrix comparing the expression of activation-dependent platelet surface analyzed directly after fixation and analyzed after 60 min of resting after fixation.

Spilover (%)						
	FL1	FL3	FL4	FL5	FL6	FL7
FL1		3.72	6.90	0.18	0.83	0.33
FL3	4.04		7.41	5.65	0.77	0.33
FL4	2.89	4.56		12.09	0.42	0.38
FL5	3.87	3.68	9.09		0.82	0.33
FL6	3.23	3.50	8.24	0,23		0.70
FL7	3.47	18.35	8.57	4.25	1.65	

FL1: SYTO13. FL3: CD42b-AF700. FL4: antifibrinogen-V420. FL5: CD45-BV650. FL6: P-selectin-PE. FL7: CD63-PECy7.

## Data Availability

The data that support the findings of this study are available from the corresponding author upon reasonable request.

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
