# Peer review of "Advanced Flow Cytometry Using the SYTO-13 Dye for the Assessment of Platelet Reactivity and Maturity in Whole Blood"

_mps, 2023, doi:10.3390/mps6010008_

Round 1
Reviewer 1 Report
In this manuscript, OBP et al describe a flow cytometry protocol for SYTO-13 labelling of platelets allowing identification and separate analysis of immature and mature platelets.
Overall the procedure will be useful for investigators of platelet reactivity in different physiological and pathological contexts.
The written description of the procedure steps and some results is sometimes confusing and hard to understand. The authors must make an effort to the easy understanding of specialists in platelet flow cytometry and general readers.
Specific points:
1. The introduction lacks the recent recommendations of the SSC- Platelet Physiology of ISTH regarding the use of flow cytometry for analysis of platelets in inherited and acquired platelet disorders; Frelinger et al; JTH 2021, PMID: 34580997
2. In 2.1 the CytoFLEX S ref is B75442, but B75408 in 2.4 (equipment)
3. In 2.3 the ref for anti-fibrinogen is missing
4. In 2.3, is it collagen or collagen-related peptide (CRP-XL from CambCOL Ltd). I guess the authors have used CRP, but collagen is mentioned throughout the paper. According to my personal experience, CRP is a very good agonist for FC assays of platelet reactivity, but not the case for collagen, and also AA
5. 3.1 subtitle does not seem appropriate. I would suggest a section entitled “storage of reagents”
6. In 3.1 correct the word antiboidies (antibodies)
7. 3.2 subtitle. I suggest “preparation of reagents”
8. 3.2.1 I guess the authors do not know the real final concentration of the commercial antibodies, and that is why they use 1:x dilutions
9. 3.2.2 the preparations of the antibody cocktail: The volumes referred correspond to the previously diluted antibodies/reagents. This should clearly specified
10. 3.3.3 Preparation of agonists. This section should be entitled "Preparations of working dilutions of agonists". Also, I don´t understand the different calculations.
For instance
- stock solution of collagen(CRP?) (section 2.3) is 7 mg/mL: 1:100 to prepare solution A and then 1:100 will give a collagen final concentration of 0.7ug/mL
- stock solution of AA is 15.3 mM: 15uL +15.6ul Hepes (1:2) giving a final concentration 7.5 mM
- ADP and TRAP calculations also seem wrong
11. 3.4.1 “collect agonist from the freezer” What does this mean? Are the agonist working solutions also stored frozen or at 4ºC?. Please clarify
12. 3.6 analysis (~90 min). What does ~90 min mean?
13. 4.3 Results; Table 1 legend, ref [20] is not listed in the reference list;
14. Note 6.2. I suggest entitling this as “Preparation of Hepes buffer” or simply “Hepes buffer”
15. Note 6.5; I guess the CytoFLEX S has four lasers but eight detectors, but this is not mentioned in the paper
16. Note 6.10 rephrase the sentence in point 6 “the analysis was performed.. “. It is hard to understand
17. Note 6.10. Maybe the conclusion should be after a better explanation of these assays shown in Fig 4&5. As they are now I don´t understand the difference
18. Note 6.12. Stability… Table 4. What is the message/conclusion of these experiments?. Are you recommending analysis immediately after fixation or following 60 min?.
19. Note 6.14. How was this compensation matrix established?. It is not mentioned at all.
Reviewer 2 Report
In the article entitled „Advanced Flow Cytometry using the SYTO-13 dye for assessment of Platelet Reactivity and Maturity in Whole Blood” by dr Pedersen et al., the Authors described and validated a novel experimental protocol. They used flow cytometric technique utilising fluorescent dye SYTO-13 having ability to stain residual RNA in blood platelets as a marker of platelet immaturity. The Authors provided very detailed study protocol and workflow. I like the idea of the novel method, besides he quality of presentation is outstanding and a paper is very well-written. However, I have a couple of points listed below that should be addressed.
1. The information regarding the dye SYTO-13 in section 2.3 Reagents (line 96) is very scarce. The name of the manufacturer is given only. Since this reagent is crucial for the study, it would be beneficial for the reader to give also a concentration of SYTO-13 and a solvent used by the manufacturer. In a case the solvent being 100% DMSO, it could potentially affect platelet function.
2. As I understand, the Authors decided to perform platelet activation with agonists and staining with a cocktails of antibodies and the dye in one, single step, followed by cell fixation. However, in the literature there are many protocols in which a step of activation with agonists is carried out prior to the step of staining. Such the approach would allow using different activation times for various agonists. I would like to know the Authors opinion on this issue.
3. Line 348: Platelet idintification -> Platelet identification
